# Large scale active-learning-guided exploration for in vitro protein production optimization

Olivier Borkowski[1,5], Mathilde Koch[2,5], Agnès Zettor[3], Amir Pandi[2], Angelo Cardoso Batista [2], Paul Soudier[2] & Jean-Loup Faulon[1,2,4 ✉]

Lysate-based cell-free systems have become a major platform to study gene expression but batch-to-batch variation makes protein production difficult to predict. Here we describe an active learning approach to explore a combinatorial space of ~4,000,000 cell-free buffer compositions, maximizing protein production and identifying critical parameters involved in cell-free productivity. We also provide a one-step-method to achieve high quality predictions for protein production using minimal experimental effort regardless of the lysate quality.

[1] Génomique Métabolique, Genoscope, Institut François Jacob, CEA, CNRS, Univ Evry, Université Paris-Saclay, 91057 Evry, France. [2] Micalis Institute, INRAE, AgroParisTech, Université Paris-Saclay, Jouy-en-Josas, France. [3] Chemogenomic and Biological Screening Core Facility, Institut Pasteur, Department of Structural Biology and Chemistry, Center for Technological Resources and Research (C2RT), 25/28 rue du Dr Roux, 75724 Paris Cedex 15, France. [4] SYNBIOCHEM Center, Manchester Institute of Biotechnology, School of Chemistry, University of Manchester, Manchester, UK. [5]These authors contributed equally: Olivier Borkowski, Mathilde Koch. ✉email: jean-loup.faulon@inrae.fr

Cell-free systems, especially lysate-based systems, are major platforms for both prototyping of genetic circuits and understanding of fundamental processes[1–7]. They provide fast gene expression kinetics, low reaction volumes, allowing high-throughput measurements and simplified gene characterization via decoupling protein production from host physiology[8–12]. Cell-free systems could disseminate among laboratories and be standard methods for molecular biology if efficient and predictable protein productions were guaranteed. Ribosomes, native polymerases and cofactors concentrations remain arduous to control as they are provided by the lysate[13,14], making the efficiency of cell-free systems variable. A great challenge is to develop a lysate-specific optimization method for cell-free buffer composition to maximize protein production. Using a design of experiment approach, Caschera et al.[13] explored cell-free buffer compositions by varying one compound concentration at a time and obtained a 10-fold increase of protein production for in vitro ribosome assembly in a lysate-based cell-free system. Such results reveal the considerable margins of improvement of protein expression for the home-made lysate-based cell-free systems that we tested in this study.

Active learning is an artificial intelligence method that makes use of machine learning algorithms to determine the next set of experiments to be carried out while studying a given problem[15]. In the context of systems biology, it was first introduced to assign protein function using yeast deletion mutants and auxotrophic growth experiments[16] and later used to predict the effect of supplied chemicals on the subcellular localization of proteins[17]. Active learning is now being used in many fields related to biology including medicinal chemistry[18] or structural biology[19]. In the context of bioproduction optimization, Design of Experiment (DoE) methods are generally preferred over active machine learning because the training set sizes on which learning is performed are rather limited[20,21]. Because cell-free systems enable one to generate large amount of data in a short time span, we explore here the use of an active machine learning strategy to optimize and understand the impact of cell-free buffer compositions on protein production in cell-free systems.

We demonstrate that a sufficient amount of data can be obtained to train a machine learning algorithm[22,23], achieve high quality predictions and increase protein production by 34 times with our home-made lysate in comparison with the initial buffer composition. We next show that only 20 informative compositions are enough to train our machine learning models and obtain accurate predictions. This approach enables to maximize protein production on different cell lysates with minimal experimental effort.

## Results

**Combinatorial space of cell-free buffer compositions.** To study cell-free systems productivity, we developed an automatable strategy coupling an acoustic liquid handling robot (Echo 550, Labcyte, USA) and a plate reader (Infinite MF500, Tecan, USA) to measure ~4000 cell-free reactions (including controls and triplicates) and provide data to train machine learning models. The lysate was obtained by sonication and supplemented with compounds described in Fig. 1a. The reference concentrations are based on the protocol developed by the Noireaux laboratory[24] (see Methods section, we fixed the maximal Mg-glutamate and K-glutamate concentration based on an initial buffer composition optimization as described in Noireaux laboratory protocol, Supplementary Fig. 1). We fixed four concentration levels for each of the 11 compounds leading to a combinatorial space of 4,194,304 possible compositions (Fig. 1a). Protein production was measured using the fluorescence level from the expression of *sfgfp* under control of a constitutive promoter (Fig. 1b and Supplementary Table 1). In order to compare measurements between plates, we maximized a relative fluorescence level named yield hereafter (Fig. 1b). The yield is defined as the ratio of the fluorescence produced with a chosen composition divided by the fluorescence obtained with the reference composition (Fig. 1b).

**Active learning strategy to optimize buffer composition.** To explore our vast combinatorial space, we used an active learning strategy[22], combining both exploration (buffer combinations with a low prediction accuracy) and exploitation (buffer combinations predicted to maximize the yield) to increase the yield and reduce model uncertainty (Fig. 2). Each iteration started with 102 new cell-free buffer compositions to be tested. The fluorescence level

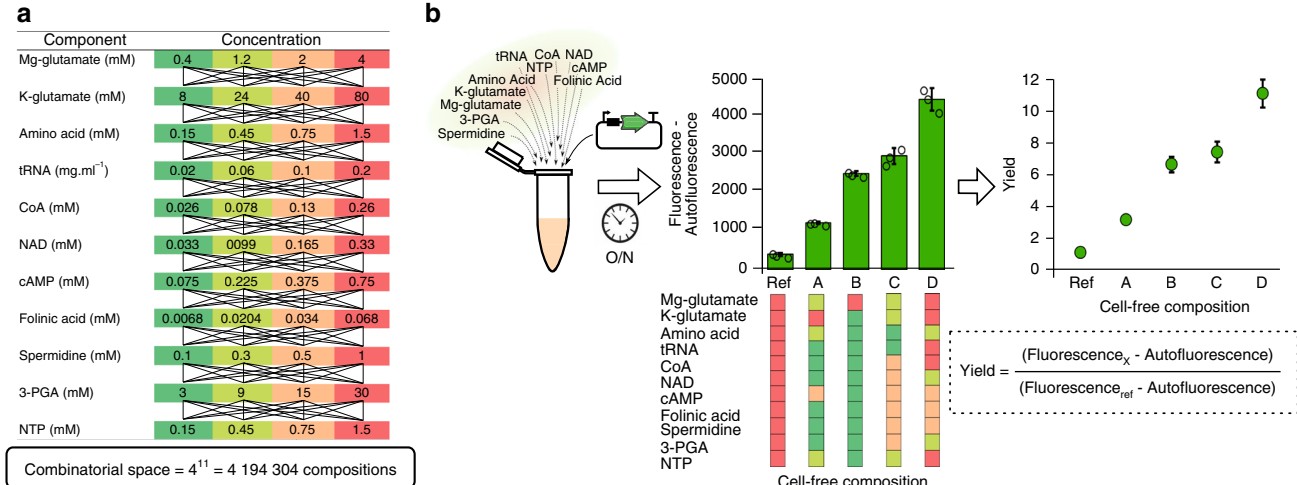

**Fig. 1 Combinatorial space of the buffer composition of a cell-free system. a** List of chemicals added to the cell-free mix in addition to PEG-8000, HEPES, and the lysate. Four concentrations have been chosen for each chemical. The concentration in red is the highest concentration, then orange, light green, dark green stand for 50, 30, and 10% of the highest concentration. **b** An example of fluorescence obtained using four cell-free compositions with our plasmid (10 nM). The autofluorescence value is measured with the reference composition without DNA and subtracted from every measurement in the plate. The yield is the ratio between the fluorescence of a composition *x* and the fluorescence of the reference composition. Data are mean values and the vertical black lines stand for the standard deviation of the three replicates (*n* = 3 independent samples). Source data are provided in the Source Data file.

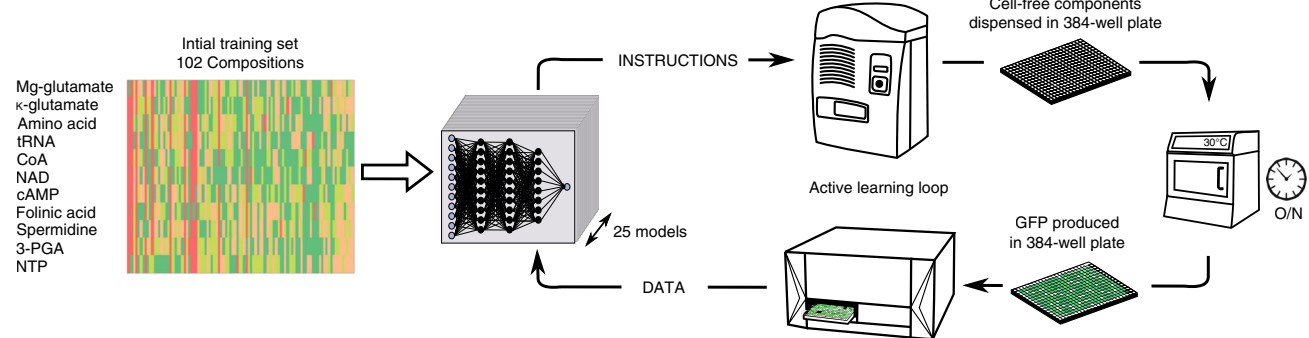

**Fig. 2 Active learning loop to explore the buffer composition of a cell-free system.** Illustration of the active learning approach used to explore the combinatorial space of cell-free composition and trained an ensemble of 25 machine learning models. Source data are provided in the Source Data file.

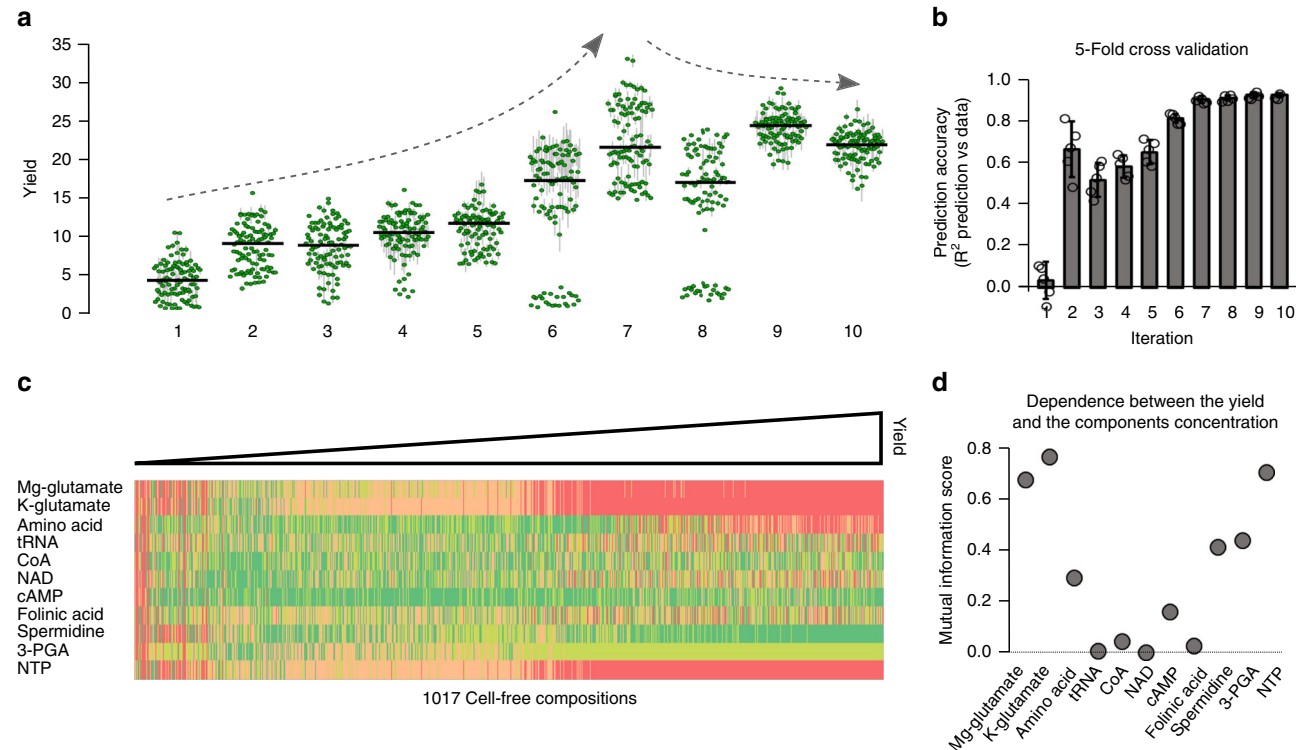

**Fig. 3 Impact of compounds concentration on protein production. a** Yield evolution amongst ten iterations. The green dots are the mean yields of three replicates obtained in the same plate with the same buffer composition ($n = 3$ independent samples). The vertical gray lines stand for the standard deviation of the three replicates. The horizontal black line is the median value of all the yields obtained during an iteration. The Arrows represent the evolution of the maximal yields value. **b** Quantification of the predictive accuracy of the model using a 5-fold cross validation. Data are mean values and the vertical black lines stand for the standard deviation of the five values ($n = 5$ scores). **c** Cell-free compositions tested in the study sorted by yield level. A row stands for one mix composition, the color code is the same as used in Fig. 1a. **d** Results of a mutual information analysis, using the 1017 compositions, of the relationship between the yield and each chemical compound. Source data are provided in the Source Data file.

was measured in a plate reader and fed to an ensemble of neural networks (Fig. 2, see Methods section). Our active learning loop was initiated with a training set of 102 cell-free buffer compositions (see methods: 22 chosen and 80 random compositions, Fig. 2). The first iteration already led to a maximum of 10-fold improvement of the yield (Fig. 3a). As expected, the initial prediction accuracy was very low (Fig. 3b). After seven iterations, we reached a maximum for both the yield (Fig. 3a) and the prediction accuracy (Fig. 3b). Eventually, we stopped at ten iterations as we were not able to increase neither the yield nor the prediction accuracy of our model (Fig. 3b, see Methods section).

Throughout our workflow, we measured fluorescence levels in 1017 cell-free buffer compositions and validated the efficiency of

our method with a high-quality predictions score ($R^2 = 0.93$) and a maximum of 34 fold increase of the yield. The 1017 cell-free compositions were sorted, from low to high yields, to observe the relationship between yield and composition (Fig. 3c). An increase of Mg-glutamate, K-glutamate, amino acids, and NTPs concentrations and a decrease of cAMP, spermidine and 3-PGA concentrations can be noticed with increasing yield (Fig. 3c). We used a mutual information analysis (see methods) to reveal the dependence between our 11 compounds concentration and the yield. Mg-glutamate, K-glutamate, amino acids, spermidine, 3-PGA, and NTPs exhibit a score between 0.25 and 0.75, confirming that a variation of their concentrations strongly impacts protein production (Fig. 3d). Variation of tRNA, CoA,

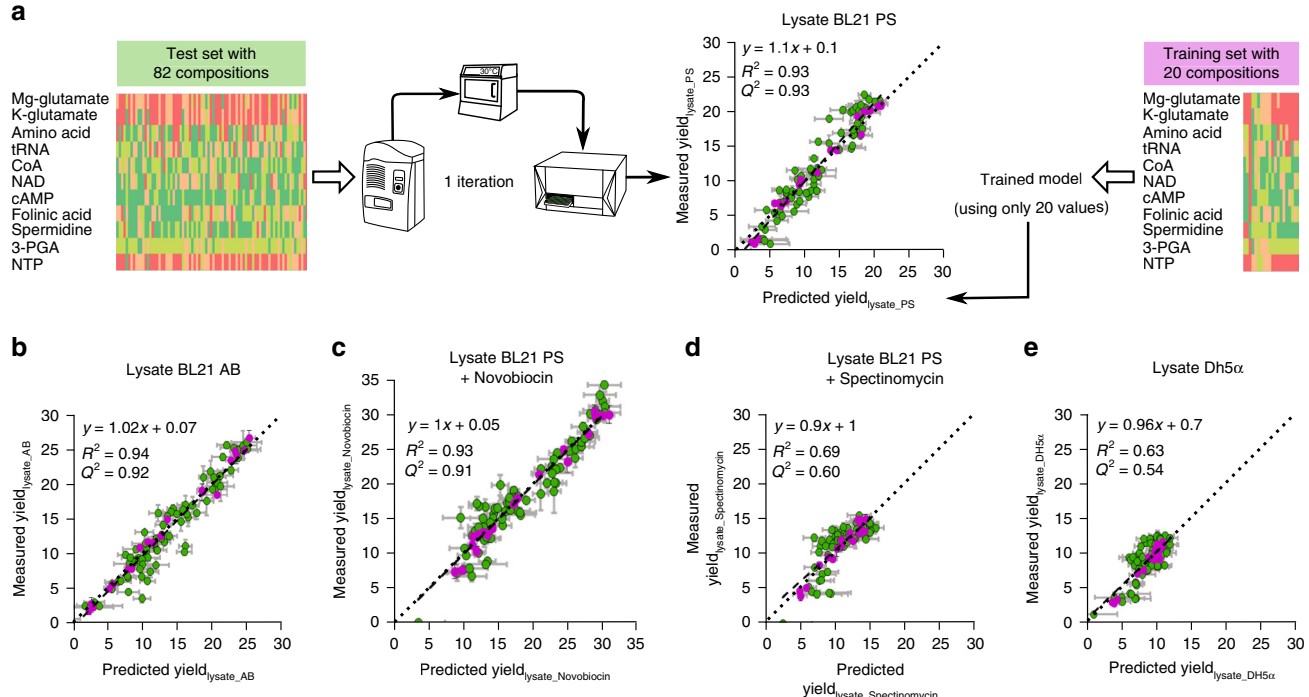

**Fig. 4 One-step method to predict protein yield in cell-free systems. a** Illustration of the method used to predict the yield of protein expression with a new lysate, labeled PS, made by another experimentalist. The training of the model is based on yield measurements of 20 buffer compositions (magenta circles). The choice of the 20 buffers combinations leading to the best predictions is described in the methods. The yields obtained with 82 compositions (green circles) were measured and compared to the model predictions to test its accuracy ($R^2$ value was computed on 102 values, $Q^2$ on the 82 values of the test set, the linear regression fits the 82 test values). The yield is specific to each lysate as the reference composition used the same chemicals concentration as in Fig. 1 but with different lysate. **b** Comparison of the yields obtained with the lysate AB (made by a third experimentalist) vs. the model predictions. **c** Comparison of the yields obtained with the lysate, of panel a, supplemented with 0.25 mg/mL of novobiocin vs. the model predictions. **d** Comparison of the yields obtained with the lysate of **a**, supplemented with 0.5 mg/mL of spectinomycin vs. the model predictions. **e** Comparison of the yields obtained with a lysate obtained from the strain DH5a vs. the model predictions. In all panels, the model predictions are based on a model trained with the same 20 buffer compositions and the same test set of 82 buffer compositions only lysate differs. In all panels, the data are mean and the horizontal gray lines stand for the standard deviation of three replicates. The horizontal gray lines stand for the standard deviation of 25 predictions and the vertical gray lines correspond to the standard deviation obtained on three replicates. Source data are provided in the Source Data file.

NAD, cAMP, and folinic acid concentrations have little impact on the yield (Fig. 3d).

**A one-step method for lysate-specific optimization.** Next, we investigated whether protein production in cell-free using lysates made in other conditions (different experimentalists, using a different strain or supplemented with antibiotics) could be quickly predicted with a one-step method (Fig. 4a). We selected 102 cell-free buffer compositions representative of the 1017 already tested with the original lysate (see Methods section, Supplementary Fig. 2a). Amongst the 102 compositions, 20 were used to train the model and 82 to test its predictive accuracy (Fig. 4a). The challenge lies in the model's ability to accurately predict a large diversity of yields based on a small training dataset. The 20 compositions (magenta dots in Fig. 2 and Supplementary Table 2) were chosen to be highly informative (see Methods section). We used the same 20 and 82 compositions to train and test our model predictions with all the lysates used in Fig. 4. With new lysates prepared by other experimentalists (labeled lysate_PS and lysate_AB), similar cell-free buffer compositions led to different yields but the compounds exhibiting a high impact on protein production remained the same (Fig. 3b and Supplementary Fig. 3). The maximum yield amongst the 102 tested compositions differs from one lysate to another, with a maximum yield at 23 and 26 for the lysate_PS and lysate_AB, respectively (Fig. 4a, b). The 102 yields obtained with the original lysate,

labeled Lysate_ORI, are presented in Supplementary Fig. 4a. The yield used previously is a relative measurement (Fig. 4 and Supplementary Fig. 5) which does not allow comparison between our cell-free systems. We calculated a global yield (calculated with the Lysate_ORI as a global reference, Supplementary Fig. 4b) and observed a maximum global yield 1.5 times higher with lysate_PS than lysate_AB (Supplementary Fig. 4c). These results highlight the variability in lysates quality even when they are prepared in the same laboratory with the same strain and protocol. Despite these differences, we achieved high quality predictions with both lysates (Fig. 4a, b). We obtained a $R^2 \sim 0.9$ for both lysates and linear fits with intercepts of 0.2/0.1 and slopes of 0.8/1.01 with lysate_PS, lysate_AB, respectively (Fig. 4a, b). These results validate our approach to both maximize protein production and accurately predict protein production regardless of the experimentalists who prepared the lysate.

We then challenged our method by interfering with the transcription or translation processes to mimic lysates of lower quality. By adding novobiocin (Fig. 4c) or spectinomycin (Fig. 4d) to the cell-free mix, we interfered with the transcription or translation apparatuses respectively. The two antibiotics led to a strong decrease in absolute protein production (Supplementary Fig. 4c) but opposite behaviors can be observed (high versus low room for yield improvement, Fig. 4c, d and Supplementary Fig. 5c, d). When the transcription process is impaired, we obtained a prediction of high accuracy with a $R^2$ of 0.91 and linear fit intercepts of 0.2 and slopes of 0.9 (Fig. 4c). The cell-free

containing novobiocin exhibits a high leeway for yield improvement (Fig. 4c and Supplementary Fig. 5c) with a maximum yield of 35 amongst the 102 cell-free buffer compositions. When the translation process is impaired, the yield is capped to a maximum improvement of 15 (Fig. 4d and Supplementary Fig. 5d). The $R^2$ value observed in Fig. 4d is lower but the linear fit exhibit an intercept of 0.1 and slopes of 0.9. Thus, we obtained accurate prediction for the low and high yields value but the intermediate yields remain difficult to estimate. Such predictions stay powerful to maximize protein production as extreme values are captured and provide precious information concerning the lysate quality (Supplementary Fig. 6, Supplementary note 2). However, a higher $R^2$ value of 0.76 is reached with this lysate when using a training set of 25 buffer compositions instead of 20 (Supplementary Fig. 7; Supplementary Table 2). Our approach exhibits efficient predictions with low quality lysates but requires more information for models training.

Eventually, we tested our method with a lysate prepared using the strain DH5α. As observed with the lysate supplemented with spectinomycin, the $R^2$ value is low but the linear fit of the data exhibits an intercept of 0.07 and slopes of 0.96. As with spectinomycin, a higher $R^2$ value of 0.80 is reached when using a training set of 25 buffer compositions instead of 20 (Supplementary Fig. 7; Supplementary Table 2). The maximum global yield obtained with this lysate was low, as expected for a strain not optimized for protein production[25] (Supplementary Fig. 4c). Nevertheless, with half of the tested cell-free buffer compositions, the Lysate_DH5α-based cell-free exhibits a high global yield (Supplementary Fig. 4c). The yield exhibits a similar behavior to the lysate supplemented with spectinomycin, suggesting an impaired translation process (little room for yield improvement, Fig. 4d, e, Supplementary Fig. 5d, e), but with a higher level of protein production.

## Discussion

Our method enables a fast lysate-specific optimization of the cell-free buffer composition to predict and maximize protein production (Fig. 4 and Supplementary Fig. 8). Our results suggest that the optimization of the cell-free buffer composition mainly improves the efficiency of the translation apparatus as we observed a limited improvement with an impaired translation. On the contrary, an inefficient transcription machinery can be balanced by the optimization of the cell-free buffer composition. Our approach gives precious information about the room for protein production improvement of a home-made cell-free system, the impact of each compound on cell-free productivity and the efficiency of the transcription and translation processes.

Eventually, we observed a pool of buffer compositions leading to high yields in all our lysates (Supplementary Figs. S4c, S5 and Supplementary Data 1). Indeed, our mutual information analysis revealed that the same compounds affect protein production in all our lysates but with a lysate-dependent sensitivity. For example, NTPs concentration highly impact protein production with Lysate_ORI but Lysate_DH5a is more sensitive to a decrease in amino acids concentration. Thus, a buffer with high NTPs and amino acids concentrations will be efficient with both lysates but using a buffer with low amino acids and high NTPs concentrations (or vice versa) will significantly decrease protein production with only one of the two lysates.

Our method, based on the measurements of GFP production with the same 20 or 25 cell-free buffer compositions used in this work to train the models provided, can be easily extended to any other bacterial-based cell-free[5,26,27] to investigate cell-free buffer optimization beyond E. coli cell-free systems. As our model is not based on mechanistic hypotheses, our method can be extended to

cell-free systems using other organisms as yeast, insect, plant or human cells after performing new explorations to find the most informative buffer compositions for those cell-systems.

## Methods

**Bacterial strains and DNA constructs.** Strains BL21 DE3 (B F– ompT gal dcm lon hsdSB(rB–mB–) λ(DE3 [lacI lacUV5-T7p07 ind1 sam7 nin5]) [malB+]K-12 (λS)) and DH5α (F– endA1 glnV44 thi-1 recA1 relA1 gyrA96 deoR nupG purB20 φ80dlacZΔM15 Δ(lacZYA-argF)U169, hsdR17(rK–mK+), λ–) were used to prepare the different lysates in this study. Our *sfgfp* plasmid was obtained by modification of the RBS of the plasmid pBEAST-J23101-sfGFP[9]. We used PCR amplifications using the reverse primer GCGGTCTCACATCTACTATTTCTCCT CTTTCTCTACTAGCTAGC and foward primer GCGGTCTCAGCTTACTTTAT CTGAGAATAGTC with the backbone, and reverse primer p CCGGTCTCAAA GCTTATCATCATTTGTACAGTTCATCC and GCGGTCTCAGATGCGTAAAG GCGAAGAG foward primer with the *sfgp* sequence. The PCR amplifications was followed by a golden gate assembly using BsaI and T4 ligase (New England Biolab) and transformed into chemically competent E. coli top10.

**Plasmid preparation.** We noticed with preliminary experiments that the same cell-compositions gave different results when we used plasmid DNA from miniprep done on different days using the same kit. The whole project was done using aliquots from the same initial batch of *sfgfp* plasmid. The plasmid was extracted from a 600 mL LB of E. coli top 10 using the Plasmid DNA purification NucleoBond Xtra Maxi of Macherey-Nagel. The 500 μL aliquots were stored at −80 °C. The whole project was done using aliquots from the same initial batch of DNA. The final *sfgfp* plasmid concentration in every reaction was 10 nM.

**Cell-free reagents preparation.** As the reagents preparation can have a significant impact on cell-free efficiency[28], all our reagents except spermidine and Mg-glutamate (we run out of those two compounds during the study) came from aliquots of the same initial batch. We did not see an impact on our control when the spermidine and Mg-glutamate were renewed.

**Cell lysate mix preparation and reactions.** The cell lysate preparation is based on the protocol of Sun et al.[24]. Briefly, the protocol of Sun et al.[24] is a 5-day protocol in three phases: harvest cells (colonies grow on plate overnight at 37 °C, 50 mL pre-culture at 37 °C for 8 h, 12 L of cultures at 37 °C until $OD_{600}$ = 1.5), lysate preparation (multiple pellet washing followed by beads-beating to obtain an lysate). The protocol was modified by using sonication instead of use of a bead beater to obtain BL21 or DH5α cell lysates. After washing the cells as following the Sun et al. protocol (Day 3 step 18) with S30A buffer (14 mM Mg-glutamate, 60 mM K-glutamate, 50 mM Tris, 2 mM DTT, pH 7.7), the cells were centrifuged 2000 × g for 8 min at 4 °C. The pellet was re-suspended in S30A (pellet mass (g) × 0.9 mL). The solution was split in 1.5 mL aliquots in 2 mL Eppendorf tubes. Eppendorf tubes were placed in a cold block and sonicated using Vibracell 72408 (from Bioblock scientific) using the following procedure: 20 s ON—1 min OFF—20 s On—1 min OFF—20 s ON. Output frequency 20 kHz, amplitude 20%.The remaining protocol followed the procedure of the Sun et al. protocol for day 3, step 37. mRNA and protein synthesis are performed by the molecular machinery present in the lysate, with no addition of external enzymes. Reactions take place in 10.5 μL volumes at 30 °C in 384-well plate. Note that we kept the 50 mM HEPES and 2% PEG-8000 fixed in every reaction. Lysate_ORI, Lysate_PS and Lysate_AB were obtained from the same E. coli strain BL21 in the same laboratory with the same sonicator and centrifuge. The Lysate_ORI came from one-batch prepared from 12 L of BL21 culture. The 12 L culture were separated in 4 L culture. The culture were inoculated, grown and their pellets were washed on different 3 days then freeze and stock at −80 °C. Then, the pellets were weighed, resuspended in S30 buffer, pooled, sonicated, centrifuged, mixed, and aliquoted on an extra day. The Lysate_PS, Lysate_AB and Lysate_PS and Lysate_DH5α were each obtained from 2 L culture. For the Lysate_spectinomycin and Lysat_novobiocin, the final concentration of novobiocin and spectinomycin were 0.25 mg/mL and 0.5 mg/mL, respectively. They were added to the cell-free reactions using Lysate_PS.

**sfGFP purification.** The sfGFP was produced in E. coli culture. After a 10 min centrifuge at 4000 × g, the pellet was resuspended in 20 mM Tris (Ph8), 0.2 M NaCl and sonicated (Output frequency 20 kHz, amplitude 40% with the Vibracell 72408). After sonication, the solution was centrifuged (4000 × g, 15 min). The proteins in the supernatant were purified and fractionated using ammonium sulfate. The sfGFP was isolated at more than 70% saturation. The solution was centrifuged (4000 × g, 15 min) and the pellet resuspended in 20 mM Tris (Ph8), 100 mM NaCl. The solution was dialyzed overnight in 20 mM Tris (Ph8), 100 mM NaCl. Eventually, for the last step of purification, we used a Mono Q anion exchange chromatography column (GE Healthcare) and obtained a solution of 90% sfGFP. The final solution dialyze in a solution 0 mM Tris (Ph8), 100 mM NaCl and 50% glycerol leading to a final concentration of 7.62 mg/mL. To obtain an absolute quantification of the protein production in cell-free, we measured the sgGFP fluorescence in wells containing 10.5 μL of sfGFP solution at different

concentration. The G-yield values are calculated as described in Supplementary Fig. 4b with the fluorescence measured from sfGFP and no autofluorescence divided by the cell-free mix lysate_ORI autofluorescence and the reference fluorescence obtained from our plasmid in a cell-free mix with lysate_ORI.

**myTXTL commercial kit**. We used the commercial kit: myTXTL from Arbor Biosciences (Sigma 70 Master Mix Kit, (USA). We used both our plasmid (10 nM final concentration) and the control plasmid, pTXTL-P70a(2)-deGFP (20 nM final concentration) provided by Arbor Biosciences. The two plasmids were expressed with the reactions provided with myTXTL kit and with the optimized cell-free reaction with the Lysate_ORI Supplementary Fig. 8b.

**Fluorescence quantification**. We used a plate reader Infinite MF500 (Tecan) to measure fluorescence in 384-well plates (Nunc 384-well optical bottom plates, Thermo-Scientific). The excitation wavelength was fixed at 425 nM, the emission at 510 nM and the gain at 50. We measured five fluorescences values for each well as a quality control of the plate reader measurements. The fluorescence was measured from the top of the 384-well plates with no lid.

**Echo liquid handler**. We used the Echo software Cherry Pick to program the Echo 550 liquid handler. The software was programmed using CSV (comma separated values) files that gave machine-readable instructions: namely the well it had to take liquid from (containing pure reagents), the well the liquid was destined to and the volume that was to be taken. It allows us to program the content of each individual well separately. We calculated the concentrations of our chemical compounds stocks so the final volumes sent to the destination well were multiples of 2.5 nL (the droplet size managed by the Echo machine). The scripts generating the CSV file are presented below in "concentrations to instructions workflow". We chose our stock volumes so that the minimal volume to transfer was 12.5 (=5 droplets).

**General script descriptions**. All scripts mentioned below were written in Python (version 3.6.5), executed in Jupyter notebooks (version 1.0.0). Scripts are available online at github (https://github.com/brsynth/active_learning_cell_free). The libraries numpy and csv were used to handle files between different scripts. We used scitkit-learn[29] (version 0.19.1) for all model training.

**Concentrations to instructions workflow**. The details of those scripts are described in the READme file of the ECHO_handling_scripts of our code. Roughly, it proceeds in four steps:

Complete concentrations: Taking as input a file containing only concentrations of interest for the machine learning algorithm, it adds information of values that are constant across all conditions, such as the lysate quantity.
Concentration to volume: This file converts a csv file concentrations -to a file of volumes one wants to test (in triplicates). This is due to the fact that the ECHO liquid handler needs volumes as inputs.
Optional: we sorted those volume files according to water content. This allows us later to manually pipet important water volumes so that the robot only adjusts small volumes.
Volume to echo: This file converts a set of transfer volume quantities to the csv file expected by the ECHO liquid handler (instructions files). It also provides a file containing the name of the wells with their corresponding transfer volumes. This file is used to match the well compositions with the fluorescence measurements obtained later with the plate reader. The amino acids and water were pipetted manually (for volumes >1 μL).
Named volumes to concentrations: maps the volumes and the associated well name to a concentration file with the associated well name, for integration with the fluorescent plate reader at the next step.

The script matched the named concentration with the yield value as described in "Data analysis" of those methods.
The chemical compounds were dispensed using BP2 fluid class except for K-glutamate and 3PGA (CP fluid class).

**Data analysis**. We provide a script to map the fluorescence quantification (see "fluorescence quantification" above) to the tested concentrations with well names (last step of "concentrations to instructions workflow" above). We performed outlier removal based on the following criteria: if the coefficient of variation, amongst three replicates, was higher than 30%, we removed the value farthest from the other 2. This concerned 27 values of our 1017 values tested during the active learning. Those are identified in the online data on Github with the third value of fluorescence is set to −1. This script also outputs csv files allowing for visualization of where the outliers are, in order to spot potential border effects. It also separately outputs the outliers for further analysis.

**Data normalization**. We normalized using the following equation:

$$\text{Yield}_{composition} = \frac{\text{Fluorescence}_{composition} - \text{Autofluorescence}}{\text{Fluorescence}_{reference} - \text{Autofluorescence}}$$

Where autofluorescence is the fluorescence measured in the cell-free reaction supplemented with water and using the reference composition. The yield exhibited in Fig. 4 used a cell-free reaction with the new lysate to measure the autofluorescence and the fluorescence with the reference composition. In supplementary Fig. 4, all the yields are calculated with autofluorescence and reference fluorescence of the Lysate_ORI.

**Quality controls**. In every 384-well plates we measured 13 control compositions (in triplicate) including the reference composition with and without DNA In each 384-well plate, we used two rows of controls: A and P. The controls in row A never changes. The controls in row P changed throughout the workflow. We used the compositions leading to higher yields in the previous iteration. When analysing our controls, we checked whether the yields were identical from plate to plate ($R^2 > 0.75$ between new plate and all previous plates on yield of controls). Plates with $R^2 > 0.75$ when compared to all previous plates, or systematically above or below other plates are discarded and the same combinations were tested again.

**Initiation of the machine learning**. For the first plate of the active learning, we proceeded as follows. We chose 22 concentrations that we wished to test: fixing all reagents at the maximum allowed concentration, except one which was at the lowest (11 combinations) and fixing all reagents at the minimum allowed concentration except one which was at the highest (11 combinations). The remaining 80 compositions were chosen randomly.

**Model training**. We used an ensemble of neural networks as performed in Caschera et al.[13] the rationale being that it easily gave us both yield and uncertainty. The models were trained as follows.
Input data is normalized: each component maximum concentration is 1, and the other values take discrete values of 0.1, 0.3, or 0.5 as described in the legend of Fig. 1. While unnormalized inputs could be used, we strongly encourage normalization due to scale differences between the inputs.
We trained an ensemble of $n$ models, where $n = 25$. For each model, training was performed ten times (models_number) using the whole dataset so far acquired (e.g., $3 \times 102$ values at the 3rd iteration). Training the model multiple times allows for optimizing for random weight initialization of the model. We kept the best model of the ten models we trained (with the highest regression from scitkit-learn $R^2$ score) and repeated the procedure 25 times to obtain an ensemble of 25 best models.
Multilayer perceptrons gave the best results (random forests and linear regressions were also investigated early on). They were trained with the default parameters from scitkit-learn except the following parameters: maximum iteration of 20,000, adaptive learning rate, adam solver, early stopping and the following layers: (10, 100,100, 20)
We obtained mean and standard error from our predictions by taking the mean and standard error from the n results generated by our ensemble of $n$ models.

**Active learning**. The workflow used the data from all the available plates as an input. It trains an ensemble of 25 models and returns instructions for the following round. Here is the detailed process:
For $N$ times ($N = 100,000$):

Randomly sample a composition in the composition space (Fig. 1a).
If a composition was drawn previously (either in a previous experiment or during current selection), neglect it.
Predict mean and standard deviation for all 100,000 points using the ensemble of 25 models previously trained.

Select the best set of compositions, according to the following Upper Confidence Bound (UCB) formula: exploitation * yield_pred_mean + exploration * yield_pred_std, with exploitation = 1 and exploration = $\sqrt{2}$. While the initial value for exploration was set to $\sqrt{2}$, which is a common value taken with UCB, if the active learning fails to improve either yield or prediction accuracy this value can easily be changed to optimize the desired characteristic. Our scripts output the best 500 compositions based on the mean and standard deviation predictions of the yields. A high standard deviation value stands for an uncertain yield value. We output compositions for full exploitation, full exploration and maximization of the above formula but use the third option for the rest of the workflow. We are therefore querying points with both high yield and uncertainty.

**Model statistics**. For model statistics presented in Fig. 3b, we used the same models as described in the active learning section above, but using 5-fold cross validation instead of the whole training set. The full dataset is separated into five subsets then the 25 models are trained on four subsets, and used to predict the 5th, where scores are obtained. This is done five times, once on each subset. The scores presented in Fig. 3b are the mean and standard deviation of those five scores.

**Mutual information calculation**. Mutual information is a method to quantify the mutual dependence between two variables. This concept is intrinsically linked to the concept of entropy and is especially useful to quantify non-linear relationships between variables. More information on the theory behind this method can be found in the review 'estimation of mutual information[30] and in sci-kit learn documentation[29]. It was calculated using the feature_selection.mutual_info_regression function from scitkit-learn[29] (version 0.19.1) between each feature and the yield (compounds_effect_analysis/mutual_information_analysis jupyter notebook) with default parameters.

**Identification of informative points**. To identify the most informative points, we proceeded in the following manner:

We did 1000 iterations of the following procedure:

Randomly sample $n$ combinations from the dataset ($n = 20$ out of a dataset of 102 values for Fig. 4).
Train models on those points using the strategy presented in model training for each lysate.
Predict on the other points (82 for Fig. 4) for each lysate.
Obtain the average score on all lysate.
Keep those combinations if this average is better.
Note: Data is saved every 100 iterations.

To improve prediction accuracy with the lysate BL21+Spectinomcyn and the lysate DH5a we used the same 20 buffer compositions as previously and 5 extra buffer compositions common to these two lysates (Supplementary Fig. 7; Supplementary Table 2). A user with a lysate of low quality can measure protein productions with this extended training set and obtain predictions of higher accuracy.

**Maximization of the protein production for future users**. Users must do the following experiments:

Maxiprep a LB culture of our plasmid (or MyTXTL plasmid).
Measure the yields (or absolute fluorescence) in the 20 cell-free compositions described in Fig. 4a.

Then, in order to apply our method to a new extract, a jupyter notebook called predict_for_new_lysate is available at https://github.com/brsynth/active_learning_cell_free. It takes as input a csv file containing tested concentrations with their corresponding yields and standard error values. It provides as an output a file to maximize exploration, exploitation, or a combination of both as in the active learning loop. For obtaining the highest possible yield, it is recommended to take the exploitation results, which contain the highest predicted yields. It must be noticed that several cell-free compositions can be predicted to reach maximum yield or values in the same range. The algorithm provides mean yields value with standard deviation errors and so several yields will be equivalent to the maximum value. During this study we provided yield values to our training algorithms but absolute fluorescence can also be used if a user does not need to compare fluorescence values measured on different 384-well plates.

**Reporting summary**. Further information on research design is available in the Nature Research Reporting Summary linked to this article.

## Data availability
All data generated and analyzed during this study are available in the Source Data file. All other relevant data are available from the authors upon reasonable request.

## Code availability
Code is available on GitHub at https://github.com/brsynth/active_learning_cell_free.

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

## Acknowledgements
O.B. is supported by Genopole "Allocation Recherche 2017" and CRI Paris "Short-term. Fellows". M.K. is supported by DGA (French Ministry of Defense) and Ecole Polytechnique. PS is supported by the ANR SynBioDiag grant number ANR-18-CE33-0015. A.C.B. acknowledge funding provided by the ANR SINAPUV grant number ANR-17-CE07-0046. A.P. is supported by INRAE (National Institute for Agricultural, Alimentation, and Environmental Research) and an idEx interdisciplinary scholarship from the University of Paris-Saclay. J-L.F. acknowledges support from BBSRC/EPSRC (grant number BB/M017702/1) and the Life Science Department of the University of Paris

Saclay. The chemogenomic and Biological Screening Platform of Pasteur is funded by the Global Care initiative and Institut Carnot Pasteur MS. We thank the Faulon Lab and Agou Lab members for fruitful discussions. We thank Claire Donnat (from Stanford University) for the data analysis advice that she kindly gave us. We thank Stephen McGovern (from INRAE) who generously provided purified sfGFP and Bikash Ranjan Semal (from INRAE) for producing Supplementary Fig. 7 predictions.

## Author contributions

O.B., M.K., and J-L.F. designed experiments. O.B. performed experiments. M.K. developed and performed model simulations and liquid handler programming. O.B. and M.K. performed data analysis. O.B. and A.Z. collected data. O.B., A.P., A.C.B., and P.S. provided lysates. A.P. cloned and maxi prep the plasmid. O.B., M.K., A.Z., and J-L.F. wrote the paper. All authors approved the manuscript.

## Competing interests

The authors declare no competing interests.
