## [Peer Review File · Nature Communications]

Reviewers' Comments:

Reviewer #1:

Remarks to the Author:

I much appreciate the authors' answers to my previous comments. The new work on the plasmids is good but not sufficient in addressing my concerns. There are no fundamental changes to the manuscript to address all three of my major comments about the limited significance, impact, and innovativeness of the work. Therefore, my recommendation remains the same as before: "The use of machine learning to improve the consistency of cell-free protein synthesis is a much-needed but not a new idea". This is a great technical work but with limited significance and innovativeness. I do not recommend the publication of the manuscript in this journal.

Reviewer #2:

Remarks to the Author:

What I wanted to suggest in the first paragraph of my previous comment is as follows:

The result of Fig. S8 experiments show that same level of products are synthesized compared to the MyTXTL kit. And this indicates that there is no considerable margins of improvement at least for the cell-free protein expression system, because the composition of MyTXTL kit may be based on the many optimization trials performed in the past. Please find my point that I'm not going to point out that the condition is not fully optimized

The authors responded that the lysate-dependent optimization of the buffer composition is required for maximizing the protein production. However, Fig. S4c and Fig. S5 clearly indicate that the highest-ranked buffer composition (No. 102 in x-axis in Fig. S4c), discovered using lysate_ORI, always resulted in almost highest protein production in different lysates and also in the reactions with inhibitors. This is highly contradictory to the authors' claim listed below.

...develop a lysate-specific optimization method for cell-free buffer composition (39-40),
...considerable margins of improvement of protein expression for most home-made lysate-based cell-free systems (43-45),
...increase protein production by 34 times with our home-made lysate (59-60),
...similar cell-free buffer compositions led to different yields (146-147),
...the variability in lysates quality even when they are prepared in the same laboratory with the same strain and protocol (156-157).

I agree that slight differences are found according to the experimental conditions and the most optimized conditions are different between the conditions. However, the difference between the most optimized condition and No. 102 condition is very subtle, which appears to be within the experimental errors.

I think that the authors also agree with my points. In Fig. S8, the authors changed the plasmid for protein expression from non-optimized one to highly-optimized one attached with the MyTXTL kit. According to the authors' claim, this experiment should be performed through the optimization of buffer composition because the translation processes in the reaction are changed from impaired

one to nearly optimal one. However, they only tested only one composition, perhaps No. 102 condition, which may suggest that authors may think that the specific composition is almost best to maximize the protein production in any conditions. I also note that I guess this specific composition may resemble with the MyTXTL buffer composition.

Thus, I suggested in my previous comment that the novelty in terms of practical results is questionable. I feel that such misleading descriptions are quite confusing to readers and therefore, if editors still would like to publish this study, I recommend to suggest that the manuscript is completely reorganized based on the finding in this study where the buffer composition for efficient protein expression is not lysate-specific.

Reviewer #3:

Remarks to the Author:

The authors have satisfactorily addressed my concerns and I also feel that they have satisfactorily addressed the primarily experimental concerns raised by the other reviewers. I think the manuscript is an important contribution and appropriate for publication in Nature Communications.

Reviewer #1 (Remarks to the Author):

I much appreciate the authors' answers to my previous comments. The new work on the plasmids is good but not sufficient in addressing my concerns. There are no fundamental changes to the manuscript to address all three of my major comments about the limited significance, impact, and innovativeness of the work. Therefore, my recommendation remains the same as before: "The use of machine learning to improve the consistency of cell-free protein synthesis is a much-needed but not a new idea". This is a great technical work but with limited significance and innovativeness. I do not recommend the publication of the manuscript in this journal.

While it is true that Design of Experiment has been used in the past with cell-free systems, to the best of our knowledge, this has not been carried out with training sets as large as the one generated in our work. Machine learning is data monger, and we show in the current manuscript how to generate large datasets, obtain remarkable prediction accuracies ($R^2 > 0.9$), and exploit these to predict protein production for new lysates. Additionally, the use of active learning (a method similar to the reinforcement learning technique used by Google Deep mind to develop the AlphaGo game), is also novel in the field of cell-free systems and in particular the exploration and exploitation strategies, which allowed us to increase protein production 34-fold in 7 iterations (Fig. 3) and find new compositions to be evaluated in different lysates to increase prediction accuracies (from $R^2=0.63$ in Figure 4 to $R^2=0.80$ in Supplementary Fig. 7 for DH5a).

Reviewer #2 (Remarks to the Author):

What I wanted to suggest in the first paragraph of my previous comment is as follows:

The result of Fig. S8 experiments show that same level of products are synthesized compared to the MyTXTL kit. And this indicates that there is no considerable margins of improvement at least for the cell-free protein expression system, because the composition of MyTXTL kit may be based on the many optimization trials performed in the past. Please find my point that I'm not going to point out that the condition is not fully optimized

We agree that, even without information about MyTXTL buffer composition, the company must have optimized its cell-free reaction. We assume that MyTXTL re-optimize its buffer composition with every new batch of lysate as we offer to do with this work.

The authors responded that the lysate-dependent optimization of the buffer composition is required for maximizing the protein production. However, Fig. S4c and Fig. S5 clearly indicate that the highest-ranked buffer composition (No. 102 in x-axis in Fig. S4c), discovered using lysate_ORI, always resulted in almost highest protein production in different lysates and also in the reactions with inhibitors.

We agree that the optimized buffer composition obtained with Lysate_ORI is an efficient composition, but not the best, for every batch (Fig. S4c, Fig. S5 and Figure R1). Nevertheless, using lysate_DH5a, we observed 20% less production in comparison with the DH5a-specific optimized composition.

Figure R1: Figure S5 with a highlight on the maximum production in reactions with 5 different lysates. The protein production in a reaction with a given lysate and the optimized Lysate_ORI buffer composition or the optimized buffer composition of this given lysate, in red and green respectively.

Symmetrically, the optimized buffer composition obtained with lysate_DH5 α performed badly with Lysate_ORI. 215% less production than the maximum (Figure R2).

Figure R2: Figure S5e with a highlight on the maximum production in reactions with lysates_ORI. The protein production in a reaction with the lysate_ORI and the optimized Lysate_ORI buffer composition or the optimized lysate_DH5 α buffer composition, in red and green respectively.

Moreover, we observed a very low correlation ($R^2=0.18$) between the Lysate_ORI and Lysate DH5a (Fig. S5e) demonstrating that the impact of the buffer composition on protein production is highly lysate_specific.

This is highly contradictory to the authors' claim listed below.

...develop a lysate-specific optimization method for cell-free buffer composition (39-40),

Our results showed that the efficiency of compositions is different with each lysate. The same components have negative, neutral or positive impacts on all lysate but with a lysate-dependent sensitivity. Thus, the best composition is different from one lysate to another.

...considerable margins of improvement of protein expression for most home-made lysate-based cell-free systems (43-45),

We obtained a 34-fold improvement with our method compared to the initial composition. We modified the line as follows: **“considerable margins of improvement of protein expression for the home-made lysate-based cell-free systems that we tested in this study.”**

...increase protein production by 34 times with our home-made lysate (59-60),

We completed this sentence with **“in comparison with the initial buffer composition”**

...similar cell-free buffer compositions led to different yields (146-147),

The order of buffer compositions is lysate-dependent. An extreme case is lysate DH5a that exhibits a very low correlation with lysate_ORI. We modified the text as follows: **“...similar cell-free buffer compositions led to different yields but the compounds exhibiting a high impact on protein production remained the same.”**

...the variability in lysates quality even when they are prepared in the same laboratory with the same strain and protocol (156-157).

Except lysate_ORI and Lysate_PS each lysate lead to different yields in our study (Fig S4c).

I agree that slight differences are found according to the experimental conditions and the most optimized conditions are different between the conditions. However, the difference between the most optimized condition and No. 102 condition is very subtle, which appears to be within the experimental errors.

To avoid any confusion for the reader, we completed our discussion to emphasize the fact that there is a pool of buffer compositions leading to high level of protein production with all our lysates even if every lysate responded differently to a change in buffer compositions:

“Eventually, we observed a pool of buffer compositions leading to high yields in all our lysates. Indeed, our mutual information analysis revealed that the same compounds affect protein production in all our lysates but with a lysate-dependent sensitivity. For example, NTPs concentration highly impact protein production with Lysate_ORI but Lysate_DH5a is more sensitive to a decrease in amino acids concentration. Thus, a buffer with high NTPs and Amino Acids concentrations will be efficient with both lysates but using a buffer with low Amino Acids and high NTPs concentrations (or vice versa) will significantly decrease protein production with only one of the two lysates.”

I think that the authors also agree with my points. In Fig. S8, the authors changed the plasmid for

protein expression from non-optimized one to highly-optimized one attached with the MyTXTL kit. According to the authors' claim, this experiment should be performed through the optimization of buffer composition because the translation processes in the reaction are changed from impaired one to nearly optimal one. However, they only tested only one composition, perhaps No. 102 condition, which may suggest that authors may think that the specific composition is almost best to maximize the protein production in any conditions. I also note that I guess this specific composition may resemble with the MyTXTL buffer composition.

In Fig S8., we tested our best result (best buffer composition with Lysate_ORI) to have an absolute comparison with myTXTL kit. We have no access to myTXTL buffer composition as the company provides reactions with lysate and buffer already mixed together. In this context, we cannot test different buffer compositions with myTXTL lysate and so cannot assume that they have the same buffer composition with similar protein production level.

Thus, I suggested in my previous comment that the novelty in terms of practical results is questionable. I feel that such misleading descriptions are quite confusing to readers and therefore, if editors still would like to publish this study, I recommend to suggest that the manuscript is completely reorganized based on the finding in this study where the buffer composition for efficient protein expression is not lysate-specific.

We completed our manuscript with additional details prompted by the reviewer comments and we added a paragraph in the discussion section regarding the pool of buffer compositions leading to high yields (see above and page 7 of the revised manuscript).

Reviewer #3 (Remarks to the Author):

The authors have satisfactorily addressed my concerns and I also feel that they have satisfactorily addressed the primarily experimental concerns raised by the other reviewers. I think the manuscript is an important contribution and appropriate for publication in Nature Communications.

We thank the reviewer for these final comments